# Influence of Adhesive Layer Thickness on the Effectiveness of Reinforcing Thin-Walled Steel Beams with CFRP Tapes—A Pilot Study

**DOI:** 10.3390/ma15238365

**Published:** 2022-11-24

**Authors:** Ilona Szewczak, Malgorzata Snela, Patryk Rozylo

**Affiliations:** 1Faculty of Civil Engineering, Lublin University of Technology, 40 Nadbystrzycka Str, 20-618 Lublin, Poland; 2Faculty of Mechanical Engineering, Lublin University of Technology, Nadbystrzycka 36, 20-618 Lublin, Poland

**Keywords:** thin-walled steel beam, CFRP tapes, reinforcement method, adhesive connection

## Abstract

When reinforcing thin-walled steel members with composite tapes, two issues often overlooked in published scientific papers should be considered, namely the correct thickness of the adhesive layer and the optimum bond length of the CFRP tape. In this article, the authors focused on the first of these issues. For this purpose, eight beams with a thin-walled box cross-section and a length of 3 m were subjected to bending in a four-point scheme. Six beams were reinforced with Sika CarboDur S512 composite tape, and two beams without reinforcement were tested as reference members. Three thicknesses of the adhesive layer (SikaDur-30) were analyzed: 0.6 mm, 1.3 mm and 1.75 mm. In addition to examining the effect of the thickness of the adhesive layer on displacements and deformations of thin-walled steel members, the load value at which the composite tape peeled off was also analyzed. Numerical analyses were then carried out in Abaqus, the outcomes of which showed good agreement with the laboratory results. Both numerical and laboratory results have shown that the thickness of the adhesive layer had a minor effect on the reduction in deformation and displacement of the tested beams. At the same time, with the increase in the thickness of the adhesive layer, the value of the load at which the CFRP tapes detached from the beam surface significantly decreased.

## 1. Introduction

The intensive development of structures with thin-walled cross-sections is mainly related to technical advances in the fabrication and assembly of steel components and the desire to minimize material consumption. According to [1], the development of thin-walled structures has reduced the consumption of steel by up to 50% compared with conventional hot-rolled cross-sectional structures, reduced assembly time by up to 60% and reduced construction costs by up to 25%. Thanks to these advantages, thin-walled elements are increasingly being used not only as building envelope elements, but also as the main elements of the load-bearing structure. It should be noted that the increasing use of thin-walled steel components is also linked to the possible need to reinforce them, for example, in the event of design errors or additional external loads. Unfortunately, thin-walled steel structures are characterized by limited reinforcement possibilities. The very thin cross-sectional walls (1 mm to 3 mm) limit the possibility of welded connections, and the implementation of connections using mechanical fasteners is significantly restricted. It is, therefore, necessary to find an efficient and easy method for strengthening this type of structure. One such method may be the use of carbon fiber strips embedded in an epoxy matrix (carbon fiber-reinforced polymer (CFRP)) bonded to the beam with adhesive.

Based on the available number of publications, the use of composite materials to reinforce hot-rolled steel structures is fairly well recognized [2,3]. Unfortunately, the situation is somewhat different for cold-formed steel components.

In [4,5,6,7,8], experiments on axially pressed thin-walled steel elements with square, circular and channel sections reinforced with CFRP were described. All studies confirmed that the use of CFRP tapes allows a significant increase in the maximum load capacity of the tested elements (from 33% to more than 200%, depending on the method of CFRP sheet placement). In addition, studies on the reinforcement of flexural steel beams with sigma sections [9,10] have shown a reduction in the horizontal displacement of reinforced beams of up to 49% relative to unreinforced beams. The cited research results give cause for optimism that CFRP tapes may find wide application to strengthen thin-walled steel structures.

According to the authors of the study, to develop the optimum method for reinforcing thin-walled steel members with CFRP tapes, attention must be paid to three issues: choosing the correct anchorage length for the CFRP tapes, determining the proper adhesive layer thickness for fixing the CFRP tapes and making the correct end of the joint. In the aforementioned studies described in [9,10], the anchorage length and adhesive thickness were based on work [3] on the reinforcement of hot-rolled steel I-section beams. In [11], the effect of the shape of the adhesive joint end on the performance of the steel composite joint is discussed. No publications describing the selection of the correct anchorage length of CFRP tapes when reinforcing thin-walled steel beams were encountered during the literature study. In addition, studies on thin-walled steel beams did not address the thickness of the adhesive layer (which is important, for example, for the failure mode of the thin-walled steel beams).

Therefore, the authors of this paper decided to address this topic and conduct a pilot laboratory study to determine the optimum thickness of the adhesive layer concerning the strengthening of steel beams with a thin-walled rectangular cross-section.

## 2. Laboratory Tests

Eight thin-walled steel beams with a rectangular cross-section of 120 × 60 × 3, 3 m long, made of S235 steel, were used for the study. Two reference beams (B1R and B2R), and six beams reinforced with Sika CarboDur S512 CFRP tape, 50 mm wide, 1.2 mm thick and 171 cm long, were subjected to bending in a four-point scheme. Based on material tests, the fundamental strength parameters of the CFRP tape were determined: Young’s modulus E = 165 GPa and Poisson’s ratio ν = 0.308. The CFRP tape was bonded to the beams with SikaDur-30 adhesive, using three adhesive layer thicknesses: in two beams, the adhesive layer thickness was 0.65 mm (beams marked B1_0.65 and B2_0.65); in two beams, 1.3 mm (B1_1.3, B2_1.3); and in two beams, 1.75 mm (B1_1.75, B2_1.75). The essential strength characteristics of the adhesive, as stated in the manufacturer’s material sheet, are a minimum compressive strength after 7 days of 75 MPa, a compressive modulus of 9600 MPa and a minimum tensile strength after 7 days equal to 26 MPa. Basic strength characteristics of the adhesive included also a minimum shear strength of 16 MPa, a minimum peel strength after 7 days equal to 21 MPa and a shrinkage of 0.04%. The adhesive was prepared according to the manufacturer’s instructions. The strength parameters of the steel were determined on the basis of laboratory tests. The steel tensile test was carried out on five samples cut from the tested beams. The shape of the test samples and the course of the test were developed in accordance with the ISO 6892-1:2009 standard. The results obtained for individual samples were repeatable. In order to determine Young’s modulus and Poisson’s ratio, the results of tensile coupon tests were statistically processed. The Young’s modulus was 207.87 GPa and the Poisson’s ratio was 0.307. The stress–strain diagram for the samples cut from the tested beams is shown below (Figure 1).

To determine the full strength of the bonded joint, the beams were tested in a Zwick&Roel testing machine (ZwickRoell GmbH & Co. KG, Ulm, Germany) 28 days after the CFRP tape was glued. The load increment was carried out with the press piston assuming a piston extension speed of 1 mm/min and recording the force every 0.01 s. Figure 2 shows a photograph of the test stand.

The load application point and the support points of the beams were dimensioned in the diagram (Figure 3). During the test, the deflection of the beams was measured using an inductive sensor, and the deformation of the beams was measured using TENMEX TFs-10 120 Ω ± 0.2% electrofusion strain gauges. The inductive sensor (LVDT2) and strain gauges (T2) were placed at the center of the beam span, as marked in Figure 3. The dimensions shown in Figure 3 are expressed in cm.

## 3. Laboratory Test Results

Figure 4a,b show load–displacement and load–deformation diagrams. To compare the results obtained, it was decided to limit the diagrams to the load value of 26.5 kN, at which the tape was detached from one of the beams.

Since the measured values obtained were highly reproducible and the graphs presented were therefore unreadable, Table 1 was prepared to analyze the effect of the adhesive layer thickness on the displacement and strain results obtained from tests on individual beams.

By averaging the results obtained for each group of beams and comparing the results of the reference and reinforced beams, it was found that strengthening the beams with CFRP tapes allows a significant reduction in displacement and deformation of the beams tested in each case. For steel members with a 0.65 mm adhesive layer thickness, vertical displacements were reduced by 9.7% and strains measured by the T2 strain gauge by 13.2%; with a 1.3 mm adhesive layer thickness, displacements were decreased by 10.3% and strains by 16.8%; and for beams with a 1.75 mm adhesive layer thickness, reductions of 9.7% and 14.9% were achieved, respectively. As can be seen, the discrepancy between the results obtained by the different groups of reinforced beams is a maximum of 3.6% when measuring strains in the tension zone and 0.6% when measuring vertical displacements. The most favorable results were obtained in beams with an adhesive layer thickness of 1.3 mm. By far, the more critical factor appears to be the moment of tape detachment from the beam. Both reference beams failed under a load of 40.7 kN. The beams with the lowest adhesive layer thickness had the highest load values at the moment of tape release.

## 4. Numerical Simulations

Numerical simulations were conducted using the Abaqus software (2022, Dassault Systemes Simulia Corporation, Velizy Villacoublay, France). As part of the numerical studies, four original numerical models were developed using the finite element method to represent parallel experimental studies. Within the framework of the first numerical model, a steel beam profile (with geometric parameters presented at the stage of describing the subject of the study), the elements supporting the structure and the element loading the structure were modeled. The other three numerical models additionally modeled a composite tape made of carbon-epoxy composite CFRP, as well as an adhesive layer (with three different thicknesses of 0.65, 1.3 and 1.75 mm, respectively, for each of the three numerical models) which directly connected the composite tape to the bottom surface of the previously modeled steel beam profile. The above approach made it possible to analyze both the behavior of a bent steel beam without reinforcement and the behavior of bent steel beams reinforced with composite tape where three different adhesive layer thicknesses were used. 

The FEA models developed allowed a faithful representation of the actual test conditions. Between the supports and the steel beam, as well as the loading element and the steel beam, contact relations were used in the tangential and normal directions, with a friction coefficient of 0.2. For numerical models containing different adhesive layer thicknesses and composite tape, an additional “tie” connection was used between the composite tape and the adhesive layer and the adhesive layer and the adjacent bottom surface of the steel beam. The steel beam was modeled as a shell model using S4R-type finite elements (shell finite elements, with a linear shape function, having four nodes for each finite element, where all translational and rotational degrees of freedom were at each node). Non-deformable finite elements of type R3D4 were used for the elements that constitute the supports and the loading element of the steel structure. Special reference points were also defined in these components with the necessary boundary conditions shown in Figure 5. 

The composite tape was also modeled as a shell element with S4R-type elements, while the adhesive layer was described with COH3D8-type finite elements (having eight nodes in each finite element). The discrete model in the basic form, i.e., without the adhesive layer and composite tapes, consisted of 16,900 finite elements and 17,325 computational nodes, while each model containing a composite tape and adhesive layer was described by 21,175 finite elements and 25,903 nodes. The size of the finite elements describing the adhesive joint was 5 × 5 mm, and the composite tape and steel beam were described by elements of 10 × 10 mm. The size of the elements in the case of the modeled adhesive layer was therefore half the size of the adjacent components. The effect of mesh density was tested, and no significant deviation from the adopted solution was observed. More significant information regarding the numerical studies, using a similar solution to the present paper, is presented in [12]. An example of a discrete model is presented in Figure 6. 

The material parameters for the main test object, which was a steel beam with a rectangular cross-section, were described by an elastic–plastic material model, the data of which were presented in the test subject description stage. The adhesive layer and the composite tape had data adequate in comparison to the information presented in a thematically similar scientific publication [13]. Material data for the composite tape and adhesive were taken mainly from [14].

## 5. Results of Laboratory Tests and Numerical Analyses

The developed numerical model showed high compliance of the results obtained numerically with those from laboratory tests. The beam failure mode was the same in the case of laboratory tests and numerical analyses, as shown in Figure 7 and Figure 8.

The high compliance of the results was also obtained when comparing the values of the vertical deflection and the deformation in the middle of the beam span, in the tension flange and in the place where the T2 strain gauge was placed. Examples of diagrams are presented in Figure 9. Figure 9 is limited to presenting the results obtained for two beams due to the legibility of the diagrams. On the graphs, the beams are marked as in the description of laboratory tests (B1R, B1_0.65, etc.), while the numerical models corresponding to the individual samples are marked by adding the letter “a” (B1Ra, B1_0.65a) at the end. The results obtained indicate that the model can be used for further numerical analyses. As in the case of laboratory tests, it was found that the reinforcement of the beams with the CFRP tapes each time allowed for a significant reduction in displacement and strain of the tested beams. However, a change in the thickness of the adhesive layer has a minimal effect on the value of vertical displacements or strain, which is presented in Figure 10.

However, in the case of the numerical analyses, the key influence of the thickness of the adhesive layer on the moment of the CFRP tape detaching from the beam, and thus the loss of reinforcement, was confirmed again. In FEM, defects in the adhesive layer appeared at a load of 37.8 kN for the 0.65 mm adhesive layer, 31.35 kN for the 1.3 mm layer, and 28.04 kN for the 1.75 mm layer, which is in line with the average results obtained by individual groups of beams in laboratory tests.

In numerical simulations, the damage to the adhesive layer was observed with the use of the SDEG parameter. The scalar stiffness degradation (SDEG) represents the damage status of interface elements adopted in a cohesive element. Moreover, damage evolution in cohesive elements is tracked via the above-mentioned parameter. The SDEG variable tells about the state of damage in the element. Obviously, initially, damage initiation was based on the maximum stress criterion (MAXS), while damage evolution was based on the energy criterion. When SDEG parameter achieves a value of 1.0, it means that the bonded joint in a given area is damaged. Figure 11 shows the damage of the glued joint.

## 6. Conclusions

The conducted laboratory tests allowed the following conclusions:-Reinforcement of the beams with CFRP tapes each time reduced the value of vertical displacements and strain in the tension zone, which at the load level of 26.5 kN were reduced by up to 10.3% and 16.8%, respectively;-The most favorable results in terms of reduction in strain and displacement values were obtained for beams with an adhesive layer thickness of 1.3 mm, which is in line with the results obtained in [3]; however, the results obtained for beams with different adhesive layer thicknesses were very similar;-As the thickness of the adhesive layer increased, the load value at which the bonded joint was damaged decreased. The difference in the load values obtained for the smallest and largest adhesive layer thicknesses was 14.4 kN, and in the case of averaged results, this value was 10.95 kN. It should be noted that this difference is particularly significant when related to the maximum value of the load, which was 40.9 kN.

The outcomes of conducted numerical analyses are consistent with the laboratory test results. They showed that the use of CFRP tapes significantly reduced the level of deflection and strain in the analyzed points of individual numerical models. In turn, the change in the adhesive layer thickness had a negligible impact on the change in the level of deflections of the beam and strain in the tension beam flange. Performed numerical analyses also confirmed the influence of the thickness of the glued joint layer on the moment of failure of the joint, which was also demonstrated in numerical analyses for a different beam cross-section described in [13]. For the beams analyzed in this study, numerical analyses showed that the increase in the thickness of the adhesive layer significantly affected the decrease in the load value at which the tape detached; the difference in the load values obtained for the extreme adhesive layer thickness was 9.76 kN.

Based on preliminary laboratory tests and numerical analyses, it can be concluded that when thin-walled steel beams are reinforced with CFRP tapes, a 0.65 mm adhesive layer thickness is the most advantageous.

## Figures and Tables

**Figure 1 materials-15-08365-f001:**
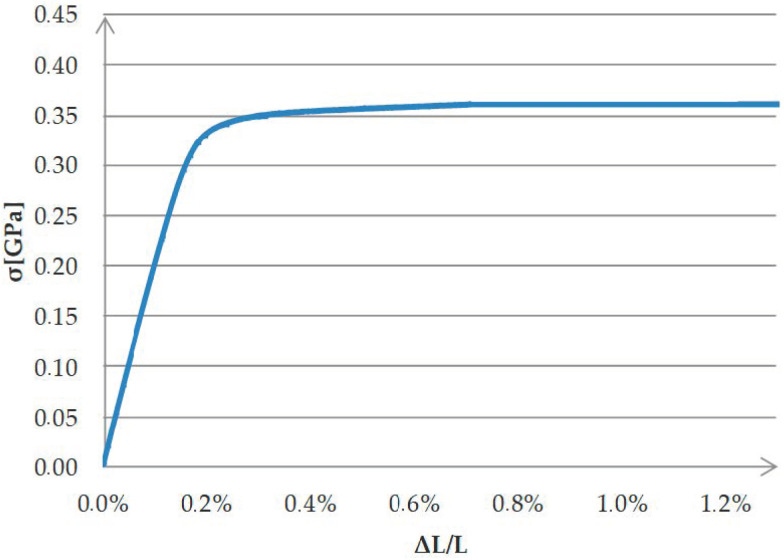
Material characteristics of steel.

**Figure 2 materials-15-08365-f002:**
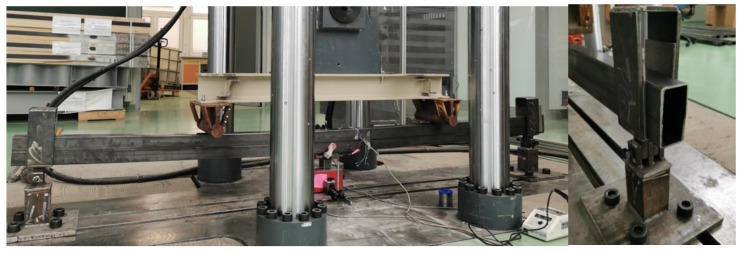
Photograph of the test stand and one of the supports.

**Figure 3 materials-15-08365-f003:**
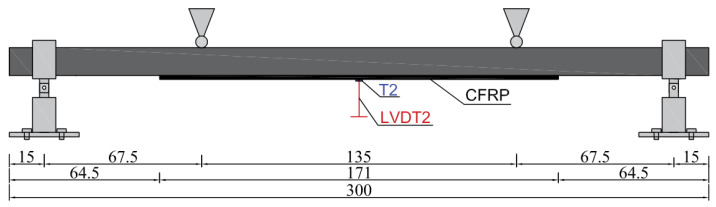
The scheme of the laboratory stand.

**Figure 4 materials-15-08365-f004:**
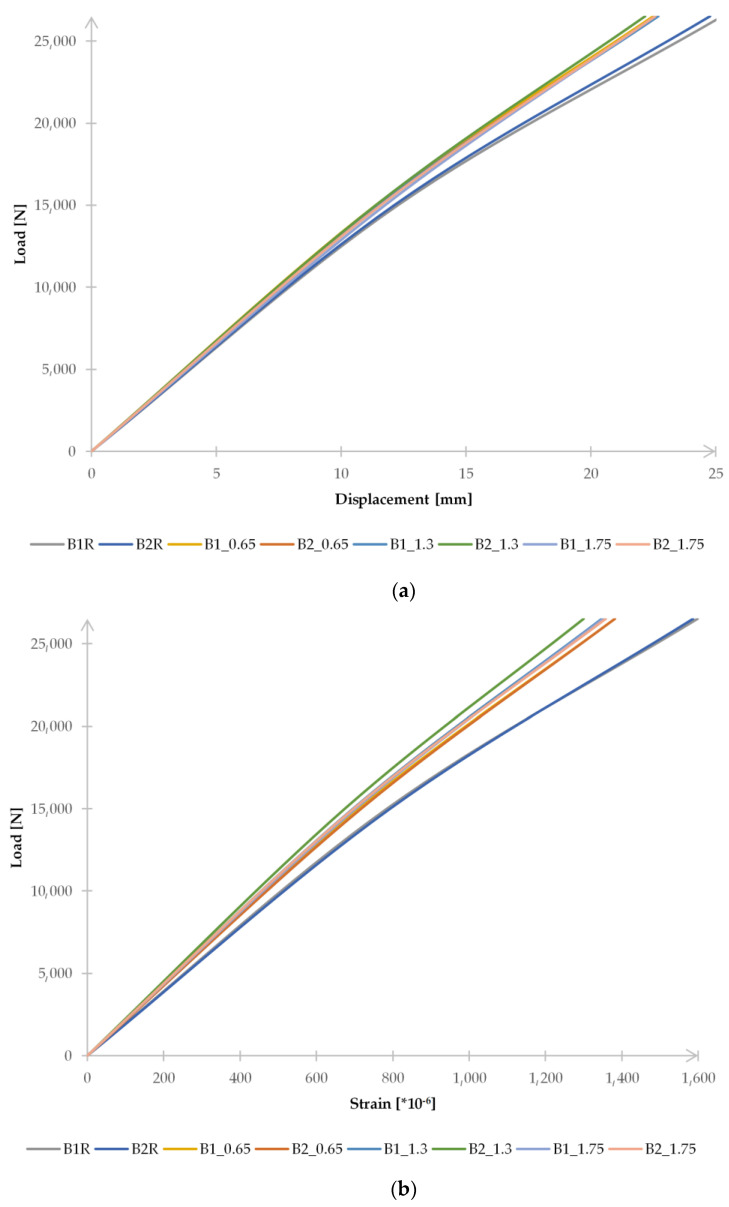
Diagram of the correlation of (**a**) vertical displacement–load at the location of the LVDT2 sensor and (**b**) deformation–load at the location of strain gauge T2.

**Figure 5 materials-15-08365-f005:**
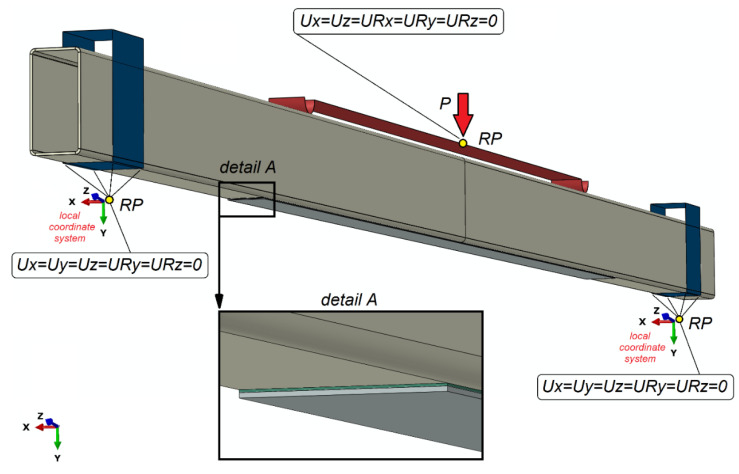
Numerical model.

**Figure 6 materials-15-08365-f006:**
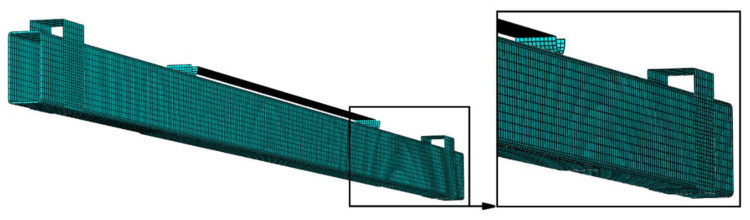
Discrete model.

**Figure 7 materials-15-08365-f007:**
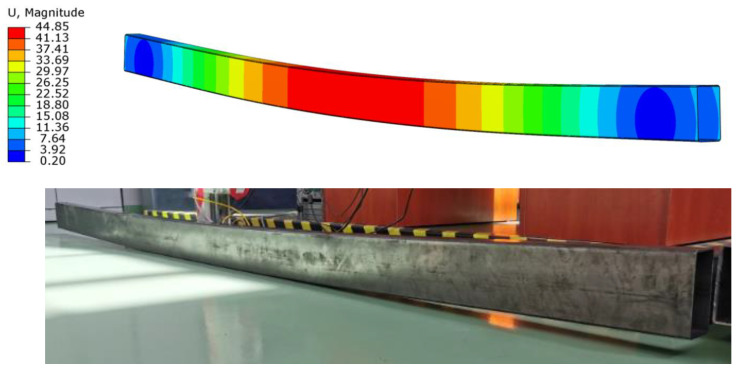
The damage of the beams obtained in numerical analyses and laboratory tests.

**Figure 8 materials-15-08365-f008:**
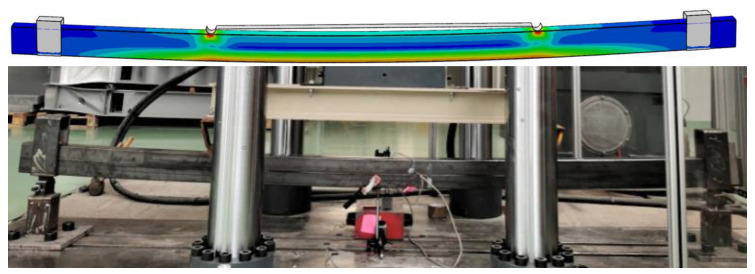
The beam failure modes obtained in numerical analyses and laboratory tests.

**Figure 9 materials-15-08365-f009:**
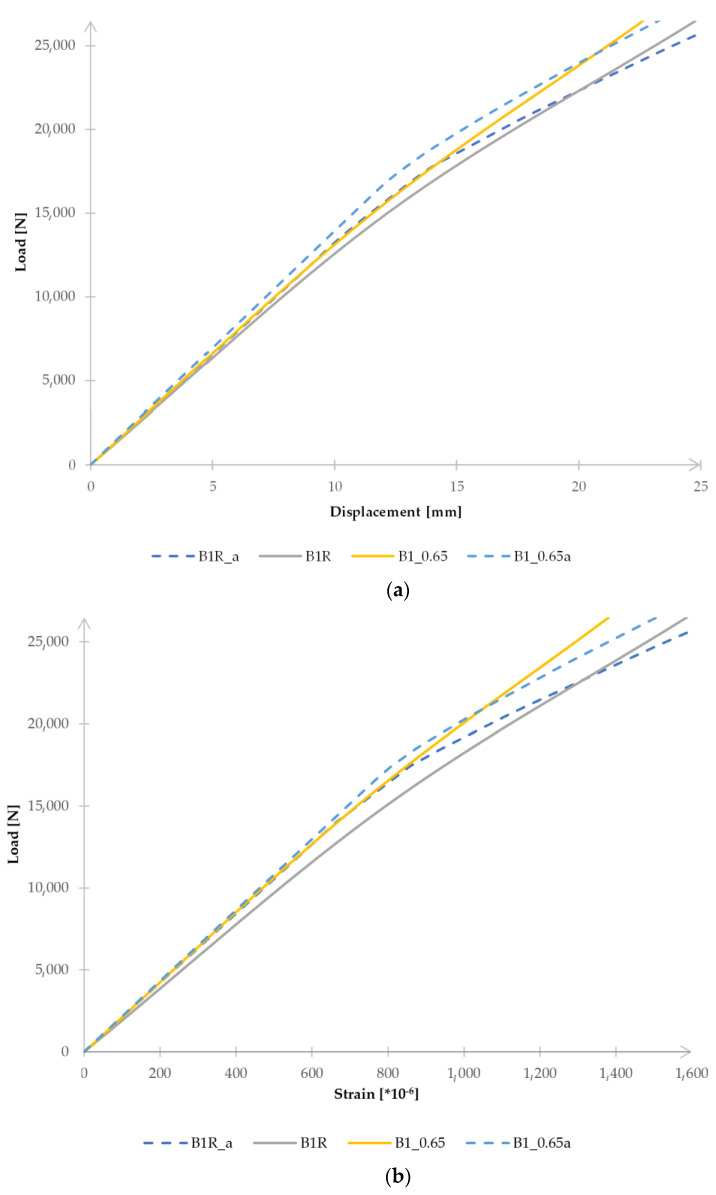
Comparison of the results of laboratory tests and numerical analyses: (**a**) vertical deflections at the location of LVDT2 sensor, (**b**) strain at the T2 strain gauge site.

**Figure 10 materials-15-08365-f010:**
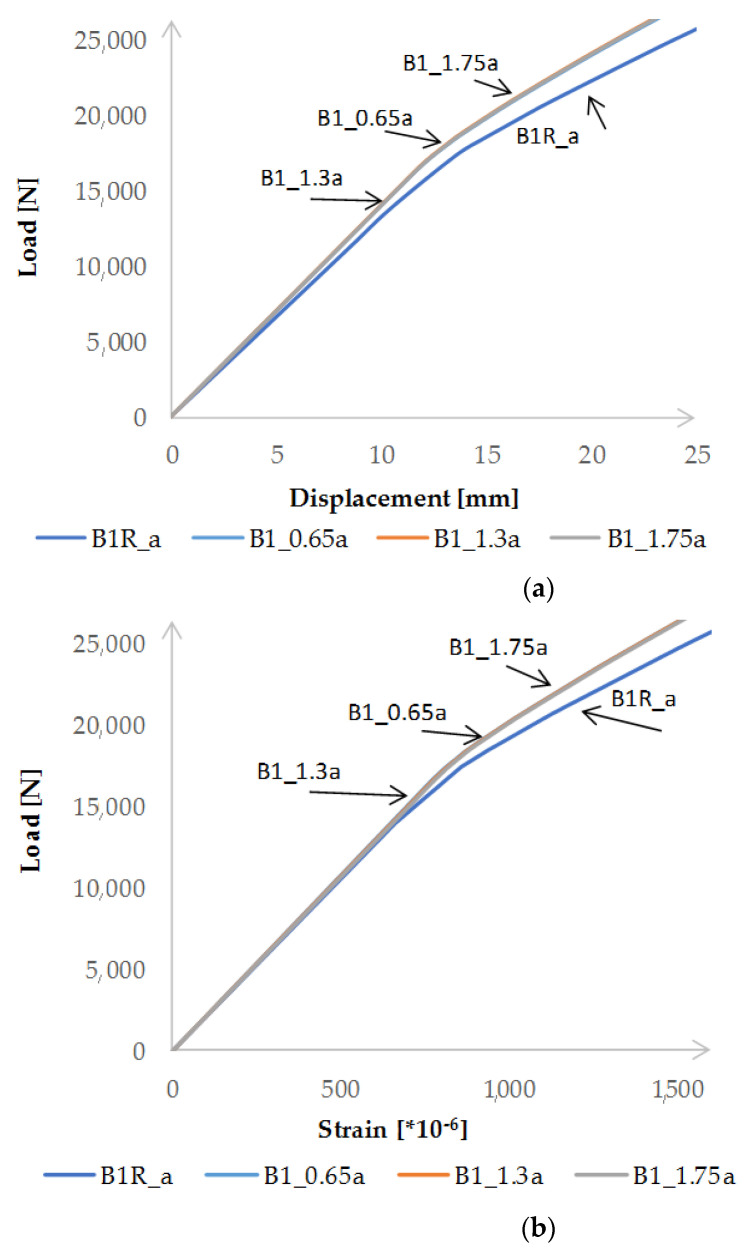
The results of numerical analyses: (**a**) vertical deflections at the LVDT2 sensor site, (**b**) strain at the T2 strain gauge site.

**Figure 11 materials-15-08365-f011:**
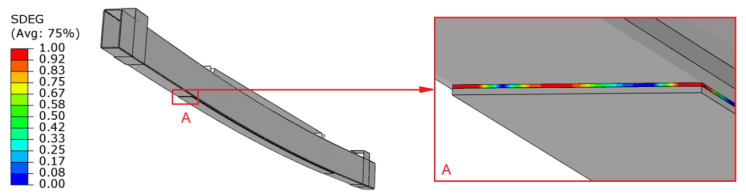
The damage of the glued (adhesive) joint.

**Table 1 materials-15-08365-t001:** Laboratory test results.

Load (kN)	Starin measured with a strain gauge T2 (×10^−6^)							
	B1R	B2R	B1_0.65	B2_0.65	B1_1.3	B2_1.3	B1_1.75	B2_1.75
26.5	1598	1587	1382	1382	1347	1300	1351	1358
Load (kN)	Vertical deflection (mm)							
	B1R	B2R	B1_0.65	B2_0.65	B1_1.3	B2_1.3	B1_1.75	B2_1.75
26.5	25.26	24.77	22.5	22.63	22.71	22.17	22.59	22.58
	Load at the CFRP tape detachment moment (kN)							
	B1R	B2R	B1_0.65	B2_0.65	B1_1.3	B2_1.3	B1_1.75	B2_1.75
	-	-	37.4	40.9	31.4	31.1	29.9	26.5

## Data Availability

Data are contained within the article.

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
