# Peer review of "Influence of Adhesive Layer Thickness on the Effectiveness of Reinforcing Thin-Walled Steel Beams with CFRP Tapes—A Pilot Study"

_materials, 2022, doi:10.3390/ma15238365_

Round 1

Reviewer 1 Report

The paper investigates the effect of reinforcing CFRP tapes on thin-walled beams, for different thickness of the adhesive layer. The issue of adhesive thickness is not new in the field of structural bonded joints and it is well known that the higher the thickness the lower the strength of the adhesive.

However, the investigation of the effect of adhesive thickness for this type of joint is sufficiently original and the work is developed following a structured scientific method.

The following issues have to be addressed.

Abstract. The last sentence, from line 22 (“The results showed…”) is not clear.

The thin-walled beam is made of S235 but the stress strain curves in figure 1 shows a yield stress of about 350 MPa. Did you perform a tensile test according to the standard? I suggest to add some details in the text.

Table 1: maybe the vertical deflection values have been confused with the strains?

Section 4. I suggest to add some detail on the FE model, in particular the side length of the elements used to describe the single lap joints. An efficient model solution for a similar structure is proposed in doi:10.1016/j.ijadhadh.2010.09.007.

Please explain what is the SDEG parameter used to identify the damage in the adhesive.

It would be interesting to assess if a lower thickness of the adhesive layer would further increase the joint strength. In addition, it is not clear how the thickness was ensured in the experimental joints.

Reviewer 2 Report

The effect of adhesive layer thickness on the effectiveness of reinforcing thin-walled steel beams with CFRP tapes is studied by combining the experiment and simulation results. It is interesting and a valuable reference for the project. Some minor repairs are needed.

1.      The abstract and conclusion should be modified and concise in order to highlight priorities and increase the reader’s interest further.

2.      Some figures should be more formal for improving the level of this manuscript.

3.      The difference between the experiment and simulation should be explained.
